# HARD MASKING FOR EXPLAINING GRAPH NEURAL NETWORKS

## ABSTRACT

Graph Neural Networks (GNNs) are a flexible and powerful family of models that build nodes' representations on irregular graph-structured data. This paper focuses on explaining or interpreting the rationale underlying a given prediction of already trained graph neural networks for the node classification task. Existing approaches for interpreting GNNs try to find subsets of important features and nodes by learning a continuous mask. Our objective is to find discrete masks that are arguably more interpretable while minimizing the expected deviation from the underlying model's prediction. We empirically show that our explanations are both more predictive and sparse. Additionally, we find that multiple diverse explanations are possible, which sufficiently explain a prediction. Finally, we analyze the explanations to find the effect of network homophily on the decision-making process of GNNs.

## 1 INTRODUCTION

Graph Neural Networks (GNNs) are a flexible and powerful family of models that build representations of nodes or edges on irregular graph-structured data and have experienced significant attention in recent years. These methods are based on the so-called "neighborhood aggregation" scheme in which a node representation is learned by aggregation of features from their neighbors and have achieved state-of-the-art performance on node and graph classification tasks. Despite their popularity, approaches investigating their interpretability have received limited attention. This paper focuses on explaining or interpreting the rationale underlying a given prediction of already trained graph neural networks.

There have been numerous approaches proposed in the literature for the general interpretability of machine learning models. The most popular approaches are feature attribution methods that intend to attribute importance to input features given an input prediction either agnostic to the model parameter (Ribeiro et al., 2018; 2016) or using model-specific attribution approaches (Xu et al., 2015; Binder et al., 2016; Sundararajan et al., 2017). However, models learned over graph-structured data have some unique challenges. Specifically, predictions on graphs are induced by a complex combination of nodes and paths of edges between them in addition to the node features. Thus explanations for a prediction should ideally be a small subgraph of the input graph and a small subset of node features that are most influential for the prediction (Ying et al., 2019).

The only existing approach for GNN explainability proposes to learn a real-valued graph mask that selects the important subgraph of the GNNs computation graph to maximize the mutual information with the GNNs prediction (Ying et al., 2019). We identify two crucial limitations of such an approach. Firstly, although mathematically tractable, a continuous mask does not ensure sparsity compared to a discrete mask – a desirable property for interpretability. Secondly, suitable notions of what constitutes an explanation in a GNN model and its evaluation are missing.

This paper proposes an alternate notion of interpretability for GNNs grounded in ideas from data compression in information theory. Specifically, we consider an explanation as a compressed form of the original feature matrix. The goodness of the explanation is measured by the expected deviation from the prediction of the underlying model. We formalize this idea of interpreting GNN decisions as an explicit optimization problem in a *rate-distortion framework*. A subgraph of the node's computational graph and its set of features are relevant for a classification decision if the expected classifier score remains nearly the same when randomizing the remaining features. This

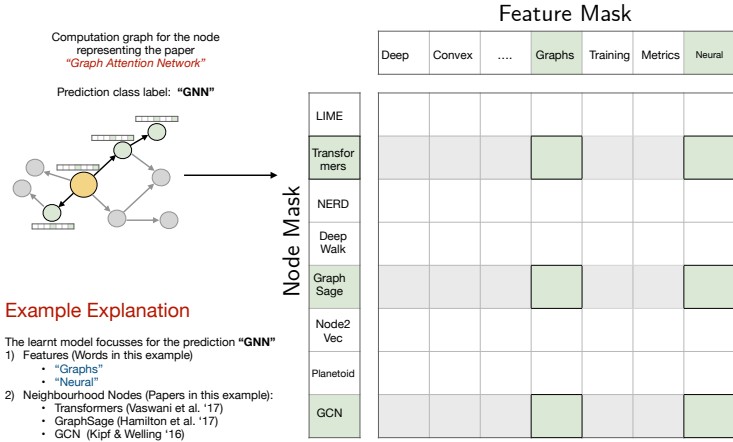

Figure 1: Computing hard masks for explaining the prediction of GNN.

formulation is arguably both a crisp, robust, and understandable notion of interpretability that is easy to evaluate. We propose a simple combinatorial procedure ZORRO that aims to find a sparse subset of features and nodes in the computational graph while adhering to a user-specified level of fidelity. Our method aims to find multiple disjoint explanations (whenever possible) that guarantee an acceptable lower bound on *fidelity to the model's decision*.

Another key problem in post-hoc interpretability of GNNs is that of evaluating explanation methods. Current evaluation methods, such as those used by GNNEXPLAINER, are primarily anecdotal and lack principled metrics. Secondly, especially for real-world datasets, there is no ground truth for the explanation, making comparison difficult. We, on the other hand, posit that an explanation is faithful to the underlying model if it retains enough predictive power – a crisp and measurable quantity. To this extent, our optimization metric, fidelity, encodes an information-theoretic interpretation of explanation – if the explanation is highly predictive in expectation, then it is a high qualitative explanation.

We conducted extensive experimentation on three datasets and four diverse GNN approaches – Graph Convolution Networks (Kipf & Welling, 2017), Graph Attention Networks (Veličković et al., 2018), GIN (Xu et al., 2019), and APPNP (Klicpera et al., 2019). Our main key findings are as follows.

1. We show that not one but multiple diverse explanations are possible that sufficiently explain a prediction. This multiplicity of explanations indicates the possible configurations that could be utilized by the model to arrive at a decision.

2. Unlike earlier mutual-information preserving interpretability approaches, i.e. GNNEX-PLAINER (Ying et al., 2019), we show that our explanations are both more predictive and sparse. We show that even with sparser explanations, our approach contains far more predictive capacity than GNNEXPLAINER.

3. We then analyze the explanations across multiple GNN models to showcase differences between their learning behavior. We specifically show that GNN models rely heavily on homophily and that prediction errors are due inability to capture homophilic signals from their neighborhoods.

## 2 RELATED WORK

Representation learning approaches on graphs encode graph structure with or without node features into low-dimensional vector representations, using deep learning and nonlinear dimensionality reduction techniques. These representations are trained in an unsupervised (Perozzi et al., 2014; Khosla et al., 2019; Funke et al., 2020) or semi-supervised manner by using neighborhood aggregation strategies and task-based objectives (Kipf & Welling, 2017; Veličković et al., 2018).

This work focuses on the post-hoc interpretability of decisions made by semi-supervised models based on graph convolution networks for node classification tasks. Inspired by the success of convolutional neural networks, graph convolution network (GCN)(Kipf & Welling, 2017) generalizes the convolution operation for irregular graph data. GCN and several of its variants follow a neighborhood aggregation strategy where they compute a node's representation by recursive aggregation and transformation of feature representations of its neighbors. For the node classification task, the final node representations are then used to predict unlabelled nodes' classes.

**Interpretability in Machine Learning.** Post-hoc approaches to model interpretability are popularized by *feature attribution* methods that aim to assign importance to input features given a prediction either agnostic to the model parameters (Ribeiro et al., 2018; 2016) or using model specific attribution approaches (Xu et al., 2015; Binder et al., 2016; Sundararajan et al., 2017). *Instance-wise feature selection* (IFS) approaches (Chen et al., 2018; Carter et al., 2018; Yoon et al., 2018), on the other hand, focuses on finding a *sufficient* feature subset or explanation that leads to little or no degradation of the prediction accuracy when other features are masked. The advantage of this formulation is that the output explanation has a precise meaning in terms of the predictive power of the chosen subset. Applying these works directly for graph models is infeasible due to the complex form of explanation, which should consider the complex association among nodes in addition to the input features.

**Interpretability in GNNs.** Model agnostic approaches like ours to interpretability in GNNs include GNNEXPLAINER (Ying et al., 2019) and XGNN(Yuan et al., 2020). GNNEXPLAINER learns a real-valued graph mask and feature mask such that the mutual information with GNN's predictions is maximized. XGNN proposed a reinforcement learning-based graph generation approach to generate explanations for the predicted class for a graph. We instead focus on explaining node level decisions. As a model introspective approach, Pope et al. (2019) extended the gradient-based saliency map methods to GCNs, which rely on propagating gradients/relevance from the output to the original model's input features. Other works (Kang et al., 2019; Idahl et al., 2019) focus on explaining unsupervised network representations, which is out of scope for the current work.

## 3  PROBLEM DEFINITION AND APPROACH

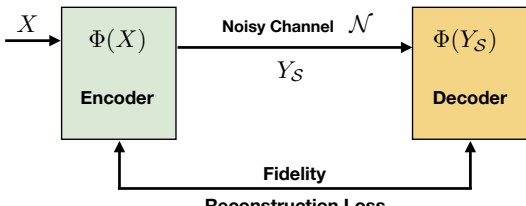

Figure 2: Our approach is based on the rate-distortion framework, which assumes two parties, the encoder and decoder. The encoder has the original (restricted) feature matrix $X$ and sends it through a noisy channel to the decoder, which receives $Y_{\mathcal{S}}$. The goal is to determine, which parts of $Y_{\mathcal{S}}$ have to have the original values from $X$, such that the fidelity is high, i.e., the decoder predicts the same label as the encoder with a high probability.

### 3.1  BACKGROUND ON GNNS

Let $G = (V, E)$ be a graph where each node is associated with $d$ dimensional input feature vector. Graph neural networks compute node representations by recursive aggregation and transformation of feature representations of its neighbors which are finally used for label prediction. Formally for a $L$-layer GNN, let $\boldsymbol{x}_n^{(\ell)}$ denote the feature representation of node $n \in V$ at a layer $\ell \in L$ and $\mathcal{N}_n$ denotes the set of its 1-hop neighbors. $\boldsymbol{x}_n^{(0)}$ corresponds to the input feature vector of $n$. The $\ell$-th layer of a GNN can then be described as an aggregation of node features from the previous layer followed by a transformation operation.

$$\boldsymbol{z}_n^{(\ell)} = \text{AGGREGATION}^{(\ell)} \left( \left\{ \boldsymbol{x}_n^{(\ell-1)}, \left\{ \boldsymbol{x}_j^{(\ell-1)} \mid j \in \mathcal{N}_n \right\} \right\} \right) \tag{1}$$

$$\boldsymbol{x}_n^{(\ell)} = \text{TRANSFORMATION}^{(\ell)}\left(\boldsymbol{z}_n^{(\ell)}\right) \tag{2}$$

Each GNN defines its own aggregation function which is differentiable and usually a permutation invariant function. The transformation operation is usually a non-linear transformation employing ReLU non-linear activation. The final node's embedding $z_n^{(L)}$ is then used to make the predictions

$$\Phi(n) \leftarrow \text{argmax}\,\sigma(\boldsymbol{z}_n^{(L)}\mathbf{W}), \tag{3}$$

where $\sigma$ is a sigmoid or softmax function depending on whether the node belongs to multiple or a single class. and $\mathbf{W}$ is a learnable weight matrix. The $i$th element of $\boldsymbol{z}_n^{(L)}\mathbf{W}$ corresponds to the (predicted) probability that node $n$ is assigned to some class $i$.

## 3.2 PROBLEM FORMULATION

We are interested in explaining the prediction of a GNN $\Phi(n)$ for any node $n$. We note that for a particular node, $n$ the subgraph taking part in the computation of neighborhood aggregation operation, see Eq. (1), fully determines the information used by GNN to predict its class. In particular, for a $L$-layer GNN, this subgraph would be the graph induced on nodes in the $L$-hop neighborhood of $n$. We will call this subgraph as the *computational graph* of the query node. We would like to pint out that the term computational graph should not be confused with the computational graph of the neural network. Let $G(n) \subseteq G$ denote the computational graph of the node $n$. Let $X(n)$, or briefly $X$ denotes the feature matrix restricted to the nodes of $G(n)$, where each row corresponds to a $d$-dimensional feature vector of the corresponding node in the computational graph.

We formulate the task of explaining the model prediction for a node $n$, as finding a partition of the components of its computational graph into a subset, $\mathcal{S}$ of relevant nodes and features, and its complement $\mathcal{S}^c$ of non-relevant components. In particular, the subset $\mathcal{S}$ should be such that fixing its value to the true values already determines the model output for almost all possible assignments to the non-relevant subset $\mathcal{S}^c$. The subset $\mathcal{S}$ is then returned as an explanation. To quantify *relevance*, we compute the expected value of *fidelity* in model's prediction for the noisy assignment to the non-relevant components.

Let us denote with $Y_{\mathcal{S}}$ the new perturbed feature matrix obtained by fixing the components of the $\mathcal{S}$ to their actual values and otherwise noisy entries. The values of components in $\mathcal{S}^c$ are then drawn from some noisy distribution, $\mathcal{N}$. Let $\mathcal{S} = \{V_s, F_s\}$ be the explanation with selected nodes $V_s$ and selected features $F_s$. Let $S$ be the mask matrix such that each element $S_{i,j} = 1$ if and only if $i$th node (in $G(n)$) and $j$th feature are included in sets $V_s$ and $F_s$ respectively and 0 otherwise.

$$Y_{\mathcal{S}} = X \odot S + Z \odot (\mathbb{1} - S), Z \sim \mathcal{N}, \tag{4}$$

where $\odot$ denotes an element-wise multiplication, and $\mathbb{1}$ a matrix of ones with the corresponding size. Figure 1 shows how the fixed elements are selected by $F_s$ and $V_s$.

**Definition.** *The fidelity of explanation $\mathcal{S}$ with respect to the graph neural network $\Phi$ and the noise distribution $\mathcal{N}$ is given by*

$$\mathcal{F}(\mathcal{S}) = \mathbb{E}_{Y_{\mathcal{S}}|Z \sim \mathcal{N}}\left[\mathbb{1}_{\Phi(X)=\Phi(Y_{\mathcal{S}})}\right]. \tag{5}$$

By fixing the fidelity to a certain user-defined threshold, say $\tau$, we are then interested in all possible disjoint sets of explanations that would have the fidelity of at least $\tau$. More precisely, our resulting set of explanations $R$ is given as

$$R = \left\{\mathcal{S}_1, \mathcal{S}_2, \ldots \mid \forall_i \mathcal{F}(\mathcal{S}_i) \geq \tau \text{ and } \underset{i}{\cap} \mathcal{S}_i = \emptyset\right\} \tag{6}$$

### CONNECTION TO THE RATE-DISTORTION THEORY

Our problem formulation is inspired by *rate-distortion theory* (Sims, 2016) which addresses the problem of determining the minimal information of a source signal that should be communicated over a leaky channel so that the source (input signal) can be approximately reconstructed at the receiver (output signal) without exceeding an expected distortion $\mathcal{D}$. In our problem, we are interested in finding a small subset $\mathcal{S}$ such that having knowledge only about the signal on $\mathcal{S}$ and filling in the rest of the information randomly will almost surely preserve the class prediction if our

chosen subset contains the information that is relevant for the model's decision. Rather than measuring distortion or disagreement in the model's decisions, we instead measure fidelity or agreement among the model's decisions with the original and the distorted signal, respectively. Distortion can be computed using fidelity as $\mathcal{D} = 1 - \mathcal{F}$. A schematic representation of our problem in terms of rate-distortion framework is shown in Figure 2.

### 3.3 OUR APPROACH: ZORRO

We propose a simple but effective greedy combinatorial approach, which we call ZORRO, to find the set of disjoint explanations with a desired level of fidelity. The pseudocode is provided in Algorithm 1. Let for any node $n$, $V_n$ denote the vertices in its computational graph $G(n)$ and $F$ denote the complete set of features. We start with zero-sized explanations and select as first element

$$\operatorname*{argmax}_{f \in F} \mathcal{F}(V_n, \{f\}) \quad \text{or} \quad \operatorname*{argmax}_{v \in V_n} \mathcal{F}(\{v\}, F), \tag{7}$$

whichever yields the highest fidelity value. We iteratively add new features or nodes to the explanation such that the fidelity is maximized over all evaluated choices. Let $V_p$ and $F_p$ respectively denote the set of possible candidate nodes and features that can be included in an explanation at any iteration. We save for each possible node $v \in V_p$ and feature $f \in F_p$ the ordering $R_{V_p}$ and $R_{F_p}$ given by the fidelity values $\mathcal{F}(\{v\}, F_p)$ and $\mathcal{F}(V_p, \{f\})$ respectively. To reduce the computational cost, we only evaluate each iteration the top $K$ remaining nodes and features determined by $R_{V_p}$ and $R_{F_p}$.

Once we found an explanation with the desired fidelity, we discard the chosen elements from the feature matrix $X$, i.e., we never consider them again as possible choices in computing the next explanation. We repeat the process by finding relevant selections completely disjoint from the ones already found. To ensure that disjoint elements of the feature matrix $X$ are selected, we recursively call Algorithm 3 with either remaining (not yet selected in any explanation) set of nodes or features. Finally, we return the set of explanations such that the fidelity of $\tau$ cannot be reached by using all the remaining components that are not in any explanation. For a detailed explanation of the algorithm details and the reasoning behind various design choices, we refer to Appendix A.

---

**Algorithm 1** ZORRO$(n, \tau, K)$

1: $V_n \leftarrow$ set of vertices in $G(n)$
2: $F \leftarrow$ set of node features
3: **return** GetExplanations$(\tau, K, V_n, F)$

---

**Algorithm 2** $\mathcal{F}(V_s, F_s)$

1: **for** $i = 0, \ldots,$ samples **do**
2:      Set $Y_{\{V_s, F_s\}}$, i.e. fix the selected values and otherwise retrieve random values from the respective columns of $\mathcal{X}$
3:      **if** $\Phi(Y_{\{V_s, F_s\}})$ matches the original prediction of the model **then**
4:          correct$+ = 1$
5: **return** $\frac{\text{correct}}{\text{samples}}$

---

**Algorithm 3** GetExplanations$(\tau, K, V_p, F_p)$

1: $\mathcal{S} = \emptyset, V_r = V_p, F_r = F_p, V_s = \emptyset, F_s = \emptyset$
2: $R_{V_p} \leftarrow$ list of $v \in V_p$ sorted by $\mathcal{F}(\{v\}, F_p)$
3: $R_{F_p} \leftarrow$ list of $f \in F_p$ sorted by $\mathcal{F}(V_p, \{f\})$
4: Add maximal element to $V_s$ or $F_s$ as in (7)
5: **while** $\mathcal{F}(V_s, F_s) \geq \tau$ **do**
6:      $\tilde{V}_s = V_s \cup \operatorname*{argmax}_{v \in \operatorname{top}_K(V_r)} \mathcal{F}(\{v\} \cup V_s, F_s)$
7:      $\tilde{F}_s = F_s \cup \operatorname*{argmax}_{f \in \operatorname{top}_K(F_r)} \mathcal{F}(V_s, \{f\} \cup F_s)$
8:      **if** $\mathcal{F}(\tilde{V}_s, F_s) \leq \mathcal{F}(V_s, \tilde{F}_s)$ **then**
9:          $F_r = F_r \setminus \{f\}, F_s = \tilde{F}_s$
10:      **else**
11:          $V_r = V_r \setminus \{v\}, V_s = \tilde{V}_s$
12: $\mathcal{S} = \mathcal{S} \cup \{V_s, F_s\}$
13: $\mathcal{S} = \mathcal{S} \cup$ GetExplanations$(\tau, K, V_p, F_r)$
14: $\mathcal{S} = \mathcal{S} \cup$ GetExplanations$(\tau, K, V_r, F_p)$
15: **return** $\mathcal{S}$

---

The pseudocode to compute fidelity is provided in Algorithm 2. Specifically we generate the obfuscated instance for a given explanation $\mathcal{S} = \{V_s, F_s\}$, $Y_{\mathcal{S}}$ by setting the feature values for selected node-set $V_s$ corresponding to selected features in $F_s$ to their true values. Figure 1 visualizes how the fixed elements of the feature matrix are determined by choice of node mask $V_s$ and feature mask $F_s$. To set the irrelevant values, we randomly choose a value from the set of all possible values for that particular feature in the dataset $\mathcal{X}$. To approximate the expected value in Eq. (5), we generate a finite number of samples of $Y_{\mathcal{S}}$. We then compute fidelity as the fraction of samples for which the

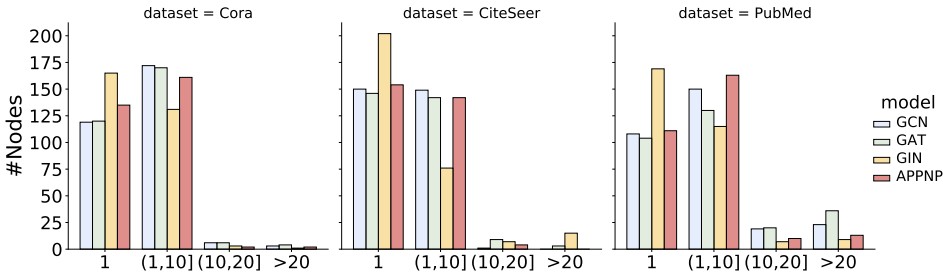

Figure 3: The number of explanations found with ZORRO at $\tau = 0.85$.

model's decision matches its original decision. Our implementation of the proposed algorithm will be made public after publication.

### CHOICE OF NOISY DISTRIBUTION $\mathcal{N}$

One might argue that the irrelevant components can be set to $0$ rather than any specific noisy value. However, this might lead to several side effects: in the case of datasets for which a feature value of $0$ is not allowed or have some specific semantics or for models with some specific pooling strategy, for example, minpool. More specifically, the idea of an irrelevant component is not that it is missing, but its value does not matter. Therefore to account for the irrelevancy of certain components given our explanation, we need to check for multiple noisy instantiations for the unselected components. Our choice of using the global distribution of features as the noisy distribution ensures that only plausible feature values are used. Next, our choice does not increase the bias towards specific values, which we would have by taking fixed values such as $0$ or averages.

## 4 EXPERIMENTS

In our experiments, we tried to answer three primary research questions:

**RQ 1.** *How often do multiple explanations exist for a given prediction? What are the sparsity-fidelity trade-offs for our approach?*

**RQ 2.** *How effective is ZORRO as compared to existing approaches in terms of fidelity and sparsity?*

**RQ 3.** *Can we discover differences in model behavior by post-hoc analysis of explanations?*

To answer these research question, we use the datasets **Cora**, **CiteSeer** and **PubMed** from Yang et al. (2016). We evaluate our approach on four different two-layer graph neural networks: graph convolutional network (**GCN**) (Kipf & Welling, 2017), graph attention network (**GAT**) (Veličković et al., 2018), the approximation of personalized propagation of neural predictions (**APPNP**) (Klicpera et al., 2019), and graph isomorphism network (**GIN**) (Xu et al., 2019). Table 5 shows the statistics of the datasets and models. For more details of our experimental setup, we refer to Appendix C.

### 4.1 MULTIPLICITY AND SIZE OF EXPLANATIONS

Our first result is that multiple (disjoint) explanations are indeed possible and are frequent. Figure 3 shows the number of nodes having multiple explanations. We observe that, without exception, all GNN models yield multiple disjoint explanations with $\approx 50\%$ of the 300 nodes under study have 2 to 10 explanations. The disjoint explanations produced by our algorithm can be understood as a disjoint piece of evidence that would lead the model to the same decision. We expect a much larger number of overlapping explanations if the restrictive condition on disjointness is relaxed. However, the objective here is to show that a decision can be reached in multiple ways, and each explanation is a practical realization of a possible combination of nodes and features that constitutes a decision. We are the first to establish the multiplicity of explanations for model predictions, unlike Ying et al. (2019) that outputs only one explanation as a soft mask over features and edges.

In general, shorter or sparser explanations are more human interpretable and hence more desirable. We conducted experiments using two fidelity thresholds ($\tau = 0.85$ and $\tau = 0.98$) and compared the

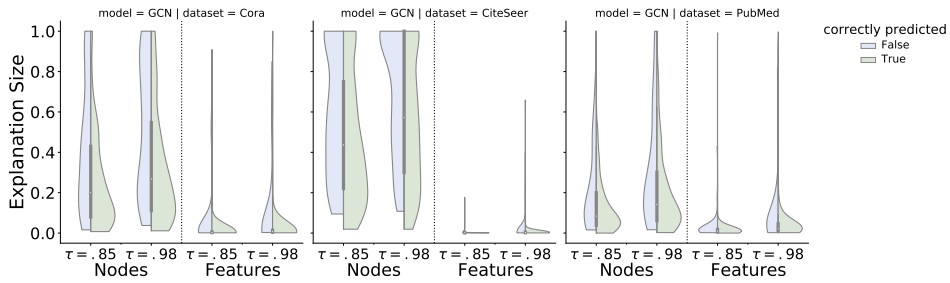

Figure 4: Comparison of ZORRO's explanation size measured by the proportion of selected elements, i.e., the number of selected nodes is divided by the number of nodes in the computational graph, and the number of selected features is divided by the number of features. For the results of all combinations, see Figure 7.

sizes of the first found explanation (see Figure 4). A key observation in this result is that ZORRO is still able to find sparse explanations for high fidelity requirements, i.e., at $\tau = 0.98$, suggesting that only a small number of nodes and features are required for most of the predictions. Although there are differences between datasets, the choice of $\tau$ has a limited influence on the explanations' size. Finally, we interestingly find that mispredictions tend to require a larger explanation size where explanation size is the fraction of nodes/features selected by ZORRO in comparison to the entire computation graph for the node.

## 4.2 COMPARISON WITH GNNEXPLAINER

We first note that unlike ZORRO an explanation from GNNEXPLAINER is a soft mask of importance scores $\in [0, 1]$ for node features and edges of the computational graph. In principle, one can choose the top-k features of GNNEXPLAINER with the highest importance values for comparing both the fidelity of both approaches. However, we choose the best settings of GNNEXPLAINER, i.e., the soft mask, to compare the fidelity distributions of both approaches (see Figure 5).

As expected, the fidelity of explanations by ZORRO for all nodes is high $[0.85, 0.95]$ or very high $(0.95, 1]$. However, GNNEXPLAINER exhibits low fidelity, i.e., $\leq 0.70$ for a fairly large $\approx 40\%$ of the nodes. The results show that our explanations are much likelier to preserve the model predictions than the soft explanations of GNNEXPLAINER.

Table 1: Average entropy $H$ of the retrieved feature masks

| method | Cora | | | | CiteSeer | | | | PubMed | | | |
|---|---|---|---|---|---|---|---|---|---|---|---|---|
| | GCN | GAT | GIN | APPNP | GCN | GAT | GIN | APPNP | GCN | GAT | GIN | APPNP |
| ZORRO ($\tau = .98$) | 2.69 | 3.07 | 4.34 | 3.18 | 2.58 | 2.60 | 4.68 | 2.78 | 2.55 | 2.58 | 3.21 | 2.86 |
| GNNEXPLAINER | 7.27 | 7.27 | 7.27 | 7.27 | 8.21 | 8.21 | 8.21 | 8.21 | 6.21 | 6.21 | 6.21 | 6.21 |

*But do the explanations from* ZORRO *have high fidelity because they are less sparse than* GNNEXPLAINER*?* To systematically measure this, we computed the entropy of normalized probability distributions over feature masks output by both approaches as a measure of sparsity, see Table 1. Note that entropy is upper bounded by the $\log$ of the number of features (see Proposition 1). The high entropy for GNNEXPLAINER corresponds to mask distribution closer to a uniform distribution, i.e., all features would have equal importance. In the case of ZORRO, the entropy is precisely equal to the $\log$ of the number of selected elements. The much lower entropy (as compared to GNNEXPLAINER) achieved by ZORRO shows that the hard masks are sparse.

**Proposition 1.** *Let $p$ be the normalized distribution of explanation (feature) masks. Then $H(p) \leq \log(|F|)$ where $M$ corresponds to complete set of features. In particular for* ZORRO *we have $H(p) = \log(selection\text{-}size)$. [Proof see Appendix B]*

We also visualize the output of the soft explanation of a node that achieves a fidelity of 1, see Figures 10 and 11. We observe that the mask values are distributed around a given value explaining the

low entropy. In such cases, when all masks for all features take low values, a small-sized explanation cannot be obtained as all components have similar importance. Effectively all features are kept in the input and no wonder that the highest value of fidelity is achieved. From the above two experiments, we conclude that ZORRO produces both sparse and high-fidelity explanations in comparison to GNNEXPLAINER.

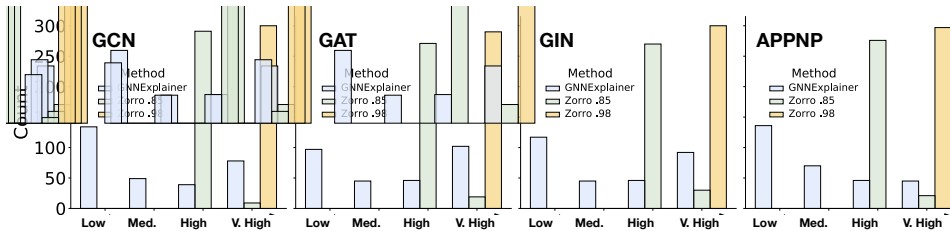

Figure 5: Comparison of fidelity, GNNEXPLAINER vs. ZORRO . **Low**: $< 0.7$, **Med.**: $[0.7, 0.85)$, **High**: $[0.85, 0.95)$ and **V. High**: $\geq 0.95$

### 4.3 HOMOPHILY OF THE EXPLANATIONS

One of the motivations of post-hoc interpretability is to use explanations to derive insights into a model's inner workings. Towards this, we investigate the homophily of the selected nodes from our explanations since GNNs are known to exploit the homophily in the neighborhood to learn powerful function approximators. We define homophily of the node as the fraction of the number of its neighbors, which share the same label as the node itself. Intuitively, it should be easier to label a node with the correct label if a larger fraction of nodes in its computational graph shares its label.

In what follows, we use homophily to refer to the homophily of a node with respect to the selected nodes in its first found explanation. True/Predicted homophily refers to the case when true/predicted node labels are used. We investigate the joint distribution of true and predicted homophily exhibited by the studied node sample. The results are shown in Figure 6. We make the following observations.

**Observation 1 -** Nodes depicting the orange regions on the extreme left side of the plots are nodes that exhibited low true homophily but high predicted homophily. The class labels for such nodes are correctly predicted. However, the corresponding nodes in the explanation were assigned the wrong labels (if they were assigned the same labels as that of the particular node in question, its predicted homophily would have been increased).

**Observation 2 -** Several vertices corresponding to blue regions spread over the bottom of the plots have low predicted homophily. These nodes are incorrectly predicted, and their label differs from those predicted for the nodes in their explanation set. The surprising fact is that even though some of them have high true homophily close to 1, their predicted homophily is low. This also points to the usefulness of our found explanation in which we conclude that nodes influencing the current node do not share its label.

**Observation 3 -** We also note that for GIN and APPNP, we have some nodes with true homophily and predicted homophily close to 1 but are incorrectly predicted. This implies the node itself and the most influential nodes from its computational graph have been assigned the same label. We can conclude that the model based its decision on the right set of nodes but assigned the wrong class to the whole group.

### 4.4 RETRAINING BASED ON LOCAL MASKS

To evaluate our ZORRO independent from our proposed fidelity measure, we use our local method as a global feature selection method and study the induced performance drop. We retrieved the explanations for all training nodes of the GNN on Cora and selected the top k features, which were most often in the first explanation. Similarly, we retrieved all explanations with GNNExplainer and selected the top k features with the highest summed feature mask values. As Table 2 shows, ZORRO outperforms GNNExplainer in all cases. Selecting the 100 most important features (with respect to ZORRO), has only a minor effect ($\Delta$ 0.01) on the test accuracy compared to the training on all 1433 features.

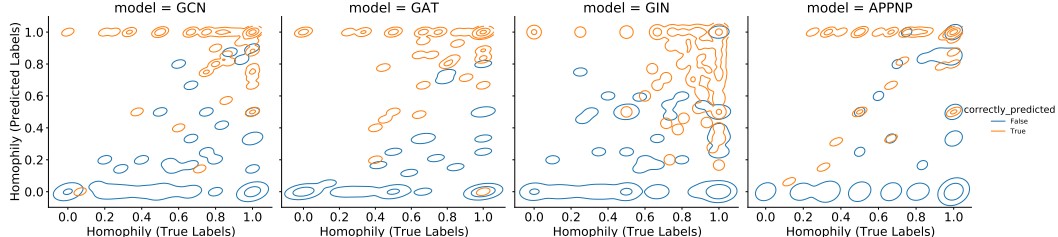

Figure 6: Dataset - PubMed. The joint distribution of the homophily with respect to the nodes selected in the ZORRO's explanation ($\tau = 0.85$) with true and predicted labels. The orange contour lines correspond to the distributions for correctly predicted nodes, and the blue one corresponds to incorrectly predicted nodes.

Table 2: Test accuracy after, retraining GCN on Cora based on the top k features. We repeated the retraining 20 times, report the mean and observed a variation of below .001 in all cases.

| Method | $k = 1$ | $k = 5$ | $k = 10$ | $k = 50$ | $k = 100$ | all |
|---|---|---|---|---|---|---|
| ZORRO ($\tau = .85$) | **0.24** | **0.50** | **0.71** | **0.77** | **0.78** | 0.79 |
| GNNEXPLAINER | 0.15 | 0.21 | 0.35 | 0.54 | 0.66 | 0.79 |

## 4.5 EXPERIMENT ON SYNTHETIC DATASET

In GNNExplainer (Ying et al., 2019), the authors proposed to use synthetic datasets built from attaching motifs, such as 'house', grid, or rings, to random Barabási–Albert (BA) graphs or regular trees. We evaluated our approach ZORRO on the synthetic dataset, which has features. The details of the experiment on the synthetic dataset are stated in Appendix E and includes some node explanations as well as the explanation of the *random closest neighborhood* baseline (in which we sample nodes randomly from the closest neighborhood). The task is to explain all the nodes from the house motif, and the ground truth are the five nodes from the corresponding house.

Table 3 shows the performance of ZORRO with $\tau = .85$ and $\tau = .98$, GNNEXPLAINER and the random closest neighborhood heuristic. The heuristic outperforms all methods with respect to recall because it selects nodes only from the closest neighborhood. However, ZORRO achieves the highest precision and accuracy and outperforms GNNEXPLAINER.

Table 3: Average performance of the node (first) explanation on the synthetic dataset.

| Method | # Nodes | Recall | Precision | Accuracy |
|---|---|---|---|---|
| Zorro ($\tau = .85$) | 2.48 | 0.35 | **0.94** | **0.90** |
| Zorro ($\tau = .98$) | 5.42 | 0.50 | 0.90 | **0.90** |
| GNNEXPLAINER | 5.34 | 0.35 | 0.33 | 0.79 |
| Random Closest Neighborhood | 5.00 | **0.67** | 0.67 | **0.90** |

## 5 CONCLUSION

We propose ZORRO as a post-hoc explanation method for the decisions made by GNN models. Inspired by rate-distortion theory, we frame the problem of explaining GNN models as a feature and node selection problem so as to minimize the expected deviation from the original decision. We proposed a simple combinatorial procedure ZORRO, which retrieves disjoint explanations consisting of *binary masks* for the features and relevant nodes while trying to optimize for fidelity. With our extensive experiments, we show multiple explanations are possible for a given decision, unlike earlier approaches that provide a soft mask. Furthermore, our explanations are sparser and achieve higher fidelity than existing approaches. Finally, our analysis of the homophily in the explanations highlighted differences in the models' behavior between correctly and wrongly predicted nodes.

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

# APPENDIX

## A   ADDITIONAL DETAILS TO OUR ALGORITHM

In the design of our ZORRO algorithm, we have made several choices, which we explicitly want to explain in detail here. In general, we have to make the following design choices: initialization of first element, iterative adding further elements, recursive design. Table 4 contains the description (which we repeat for completeness) of all variables used within the Algorithms 1, 2, and 3.

Table 4: Notation used in the algorithms

| Variable | Description |
| --- | --- |
| $n$ | Explained node |
| $\tau$ | Threshold of fidelity |
| $K$ | Number of nodes and features to evaluate per iteration |
| $V_n$ | Nodes in the computational graph of $n$ |
| $G(n)$ | Computational graph of $n$ |
| $F$ | Set of all possible features |
| $V_p$ | Set of possible nodes that can be included in an explanation |
| $F_p$ | Set of possible features that can be included in an explanation |
| $\mathcal{S}$ | Set of all explanations |
| $V_r$ | Set of remaining nodes |
| $F_r$ | Set of remaining features |
| $V_s$ | Selected nodes, i.e., nodes in the current explanation |
| $F_s$ | Selected features, i.e., features in the current explanation |
| $R_{V_p}$ | Ordered list of the possible nodes |
| $R_{F_p}$ | Ordered list of the possible features |
| $\mathcal{F}(\cdot, \cdot)$ | Fidelity, which takes a node set as first argument and a feature set as second argument |
| $\tilde{V}_s$ | Best node candidate set found |
| $\tilde{F}_s$ | Best feature candidate set found |
| $Y_{\{V_s, F_s\}}$ | Randomized feature matrix, where the features $F_s$ of the nodes $V_s$ are kept fixed, see Eq. (4) |
| $\Phi(\cdot)$ | GNN evaluated on a specific feature matrix, in Alg. 2 only evaluated to retrieve the class label of node $n$ |
| $\mathcal{X}$ | Feature matrix of all nodes in $G$ |

**Initialization of first element.** A single explanation $\{V_s, F_s\}$ consist of selected nodes $V_s$ and selected features $F_s$. The challenge to select the first node and feature is the following: Selecting

only a node or only a feature yields a non-informative value, i.e., $\mathcal{F}(\{v\}, \emptyset) = c$ and $\mathcal{F}(\emptyset, \{f\}) = c$ for all $v \in V_n$ and $f \in F$ and some constant $c \in [0, 1]$. The search for the optimal first pair would require $|V_p||F_p|$ evaluations of the fidelity, which is in most cases too expensive. Therefore, we propose to use a different strategy, which also contains information for the following iterations. Instead of evaluating, which pair of feature and node yields the highest increase, we assess the nodes and features in a maximal setting of the other. To be more precise, we assume that, if we search for the best node, all (possible) features $F_p$ were unmasked:

$$\underset{v \in V_p}{\operatorname{argmax}} \ \mathcal{F}(\{v\}, F_p) \tag{8}$$

Similarly for the features, we assume that all (possible) nodes are unmasked:

$$\underset{f \in F_p}{\operatorname{argmax}} \ \mathcal{F}(V_p, \{f\}) \tag{9}$$

Whichever of the nodes or features yields the highest value is the first element of our explanation. Consequently, the next selected element is of a different type than the first element, e.g., if we first choose a node, the next element is always that feature, which yields the highest fidelity based on that single node. We perform this initialization again for each explanation since for each explanation, the maximal sets of possible elements $V_p$ and $F_p$ are different.

**Iterative search.** The next part of our algorithm, which is the main contributor to the computational complexity, is the iterative search for additional nodes and features after the first element. A full search of all remaining nodes and features would require $|V_r| + |F_r|$ fidelity computations. To significantly reduce this amount, we limited ourselves to a fixed number $K$ nodes and features, see Algorithm 3. To systematically select the $K$ elements, we use the information retrieved in the initialization by Eq. (8) and (9). We order the remaining nodes $V_r$ and $F_p$ by their values retrieved for Eq. (8) and (9) and only evaluate the top $K$. In Algorithm 3, we have denoted these orderings by $R_{V_p}$ and $R_{F_p}$ and the retrieval of the top K remaining elements by $\operatorname{top}_K(V_r, R_{V_p})$ and $\operatorname{top}_K(F_r, R_{F_p})$. We also experimented with evaluating all remaining elements but observed no performance gain or inferior performance to the above heuristic. As a reason, we could identify that in some cases, the addition of a single element (feature or node) could not increase the achieved fidelity. Using the ordering retrieved from the "maximal setting", we enforce that those elements are still selected, which contain valuable information with a higher likelihood. In addition, we experimented with refreshing the orderings $R_{V_p}$ and $R_{F_p}$ after some iterations but observed similar issues as in the unrestricted search.

**Recursive design.** We explicitly designed our algorithm in a way such that we can retrieve multiple explanations, see line 13 and line 15 of Algorithm 3. We recursively call the Algorithm 3 twice, once with a disjoint node-set, the call in line 15 (only elements from the remaining set of nodes $V_r$ can be selected), and similarly in line 13 with a disjoint feature set. Hence, the resulting explanation selects disjoint elements from the feature matrix since either the rows or columns are different from before. As greedy and fast stop criteria, we used each further iteration, the maximal reachable fidelity of $\mathcal{F}(V_p, F_p)$.

**Complexity analysis.** The computational complexity of Algorithm 3 to retrieve an explanation $\{V_s, F_s\}$ with possible nodes $V_p$ and possible features $F_p$ is

$$O(\#\text{samples} \times (|V_p| + |F_p| + K(|V_s| + |F_s|))O(\Phi)),$$

where $O(\Phi)$ is the computational complexity for the forward pass of the GNN on the computational graph $G(n)$. For the first explanation, we have

$$O(\#\text{samples} \times (|V_n| + |F| + K(|V_s| + |F_s|))O(\Phi)).$$

# B  PROOF OF PROPOSITION 1

*Proof.* We first compute the normalized feature mask distribution, $p(f)$ for $f \in F$ ($F$ is the complete set of features). In particular, denoting the mask value of $f$ by $\operatorname{mask}(f)$, we have

$$p(f) = \frac{\operatorname{mask}(f)}{\sum_{f' \in F} \operatorname{mask}(f)}$$

Then $H(p) = -\sum_{f \in F} p(f) \log p(f)$ which achieves its maximum value for the uniform distribution, i.e., $p(f) = \frac{1}{|F|}$. For ZORRO, let $F_s$ be the set of selected features. For each $f \in F_s$, we then have $p(f) = \frac{1}{|F_s|}$ and 0 otherwise. The computed entropy is then equal to $\log(|F_s|)$. We want to point out that the proposition also follows for the case of node masks. □

## C   EXPERIMENTAL SETUP

We focus on explaining the decisions of GNN models with respect to the task of node classification. We fix the number of layers to two for all models and keep the rest of the *model architectures* and parameters as in the original paper. We train the models 200 epochs with ADAM optimizer and a learning rate of 0.01 and a weight decay of 0.0005. We use the model and GNNExplainer implementations of PyTorch Geometric Library (Fey & Lenssen, 2019). For GNNExplainer, we use the default values of 100 epochs and a learning rate of 0.01. For each dataset, we randomly selected 300 nodes and retrieved the explanations of GNNExplainer, and our method for the fidelity values $\tau \in \{0.85, 0.98\}$. We will publish the list of selected nodes together with the implementation. We used 100 samples to calculate the fidelity with Algorithm 2 and set $K = 10$ in our experiments.

Our implementation is based on PyTorch Geometric 1.6 and Python 3.7. All methods were executed on a server with 128 GB RAM and Nvidia GTX 1080Ti.

**Datasets.**   Three well-known citation network datasets **Cora**, **CiteSeer** and **PubMed** from Yang et al. (2016) where nodes represent documents and edges represent citation links. The class label is described by a similar word vector or an index of category. Statistics for these datasets can be found in Table 1. We used the datasets, including their training and test split from the PyTorch Geometric Library, which corresponds to the data published by Yang et al. (2016).

Table 5: Datasets and statistics. The test accuracy is calculated on 1000 nodes.

| Name | Classes | Features | $|V|$ | $|E|$ | Test Accuracy | | | |
|------|---------|----------|-------|-------|-----|-----|-----|-------|
| | | | | | **GCN** | **GAT** | **GIN** | **APPNP** |
| Cora | 7 | 1433 | 2708 | 10556 | 0.794 | 0.791 | 0.679 | 0.799 |
| CiteSeer | 6 | 3703 | 3327 | 9104 | 0.675 | 0.673 | 0.480 | 0.663 |
| PubMed | 3 | 500 | 19717 | 88648 | 0.782 | 0.765 | 0.590 | 0.782 |

**Models.**   As models we selected the well-known graph convolutional network (**GCN**) (Kipf & Welling, 2017) and graph attention network (**GAT**) (Veličković et al., 2018) as well as the approximation of personalized propagation of neural predictions (**APPNP**) (Klicpera et al., 2019), and graph isomorphism network (**GIN**) (Xu et al., 2019). APPNP utilizes a connection between PageRank and GCN, especially those with many layers, and extends GCNs based on the personalized PageRank. Xu et al. (2019) proposed GIN to match the representational power of the Weisfeiler-Lehman graph isomorphism test by extending the expressiveness of the feature aggregation.

## D   ADDITIONAL EXPERIMENT RESULTS

This section contains additional visualizations:

Figure 7 shows the explanation size of Cora, CiteSeer, and PubMed for the models GCN, GIN, GAT, and APPNP.

Figure 8 and Figure 9 visualize the joint homophily distributions for the dataset Cora and CiteSeer.

Figure 10 shows the feature mask distribution of GNNEXPLAINER for single nodes with fidelity 1. Figure 11 shows the feature mask distribution of GNNEXPLAINER for all explained nodes.

To be transparent about the time of executing ZORRO, Figure 12 visualizes the runtime recorded during our experiments to retrieve the first explanation for ZORRO. For the runtime experiments, we include the gradient approach used as baseline in GNNEXPLAINER: GRAD is a gradient-based

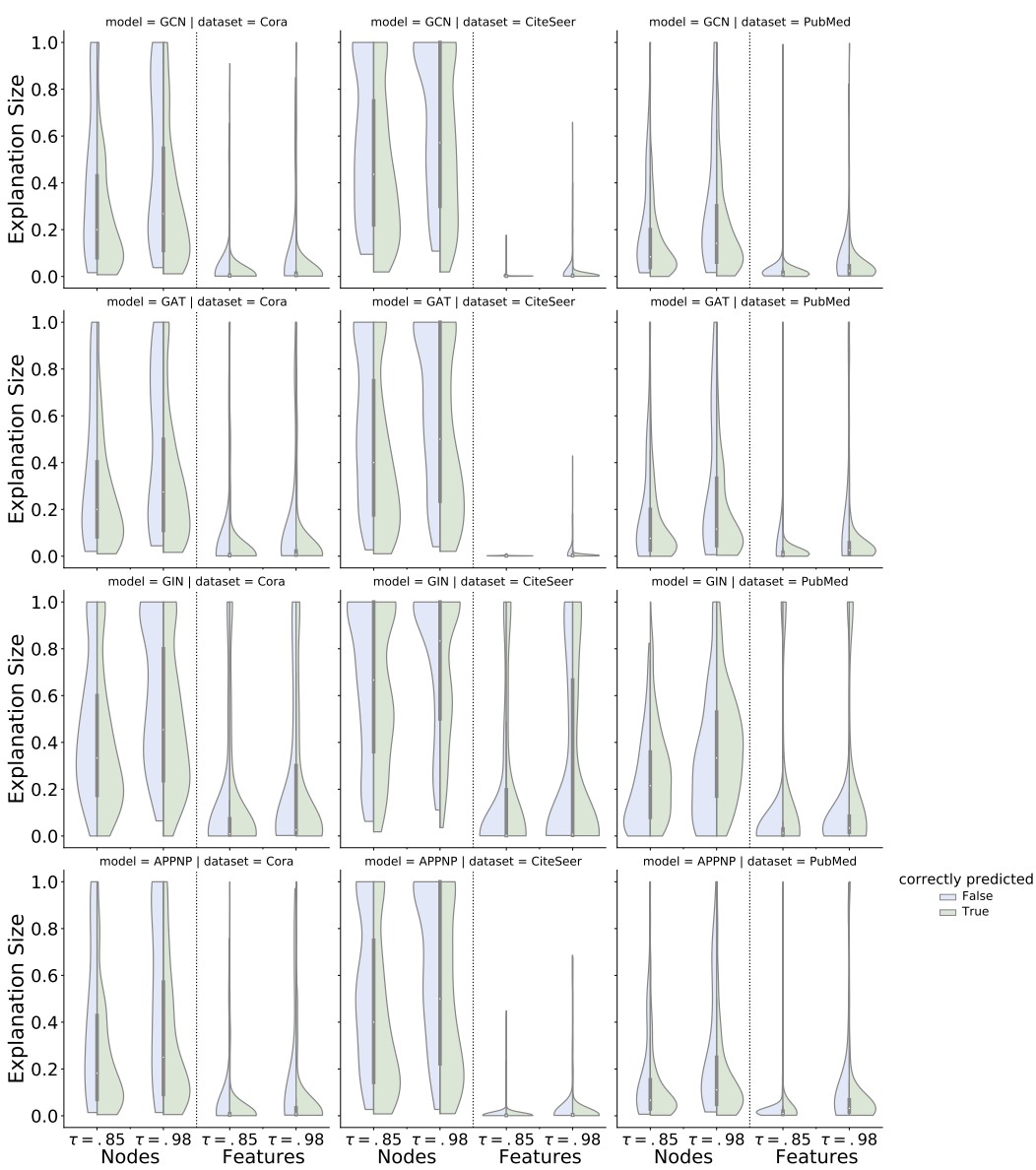

Figure 7: Comparison on ZORRO's explanation size measured by the proportion of selected elements, i.e. the number of selected nodes is divided by the number of nodes in the computational graph and the number of selected features is divided by the number of features.

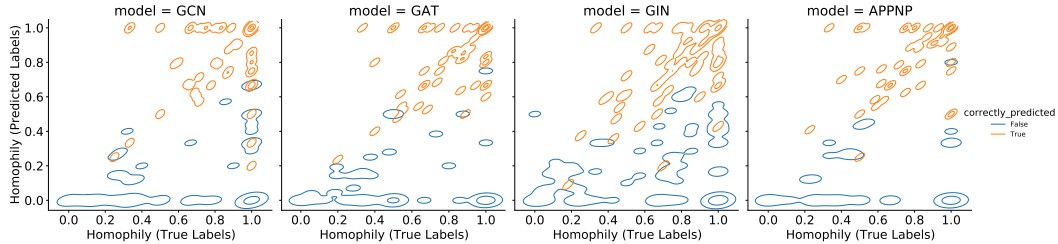

Figure 8: Dataset-Cora. Joint distribution of the homophily with respect to the selected subgraph (in the explanation of ZORRO ($\tau = .85$)) with true and predicted labels. The orange contour lines correspond to the distributions for correctly predicted nodes and the blue one correspond to incorrectly predicted nodes.

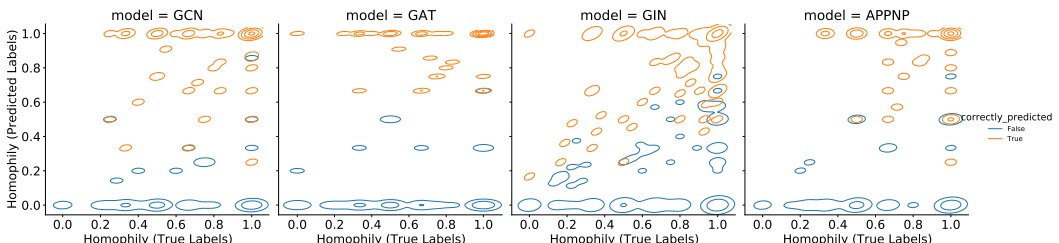

Figure 9: Dataset-CiteSeer. Joint distribution of the homophily with respect to the selected subgraph (in the explanation of ZORRO ($\tau = .85$)) with true and predicted labels. The orange contour lines correspond to the distributions for correctly predicted nodes and the blue one correspond to incorrectly predicted nodes.

method in which we compute gradient of the GNN's loss function with respect to the adjacency matrix and the associated node features (Ying et al., 2019) As stated in the computational complexity above, we note that the runtimes do not directly follow the graph size. To be precise, our approach is strictly local, i.e., it is independent of the input graph, however large it might be. The fastest average runtime we observed on PubMed, which has the highest number of nodes. Secondly, we indeed have a tunable relationship between fidelity threshold and runtime.

Currently, our implementation follows the presented Algorithms 1-2, i.e., is designed for explained a single node. If multiple explanations $\bar{V} \subset V$ are requested, this initialization step can be performed for all requested nodes at the same time (for the first explanation). Then $V_n$ has to be replaced by $\cup\{V_n : n \in \bar{V}\}$ and in each step of the fidelity, the prediction agreement with respect $\bar{V}$ has to be checked and saved separately. Hence, the orderings $R_{V_p}$ and $R_{F_p}$ are computed for all nodes

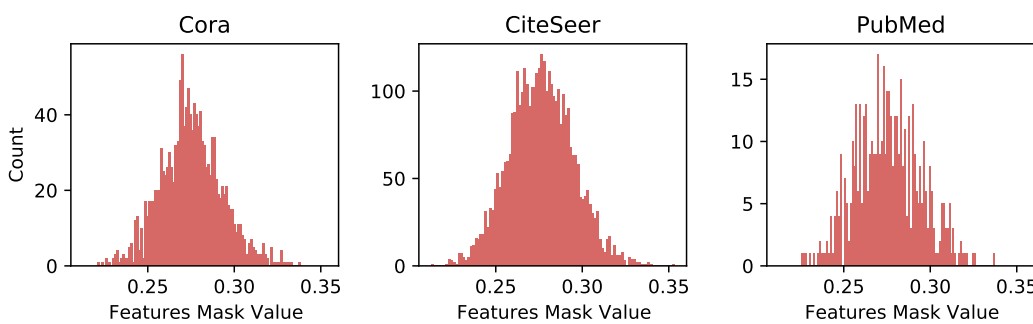

Figure 10: Distribution of GNNEXPLAINER' feature mask values corresponding to explanations of a single node of GCN, where a fidelity of $1.0$ is achieved. For the distribution of all nodes, models, and datasets, we refer to Figure 11.

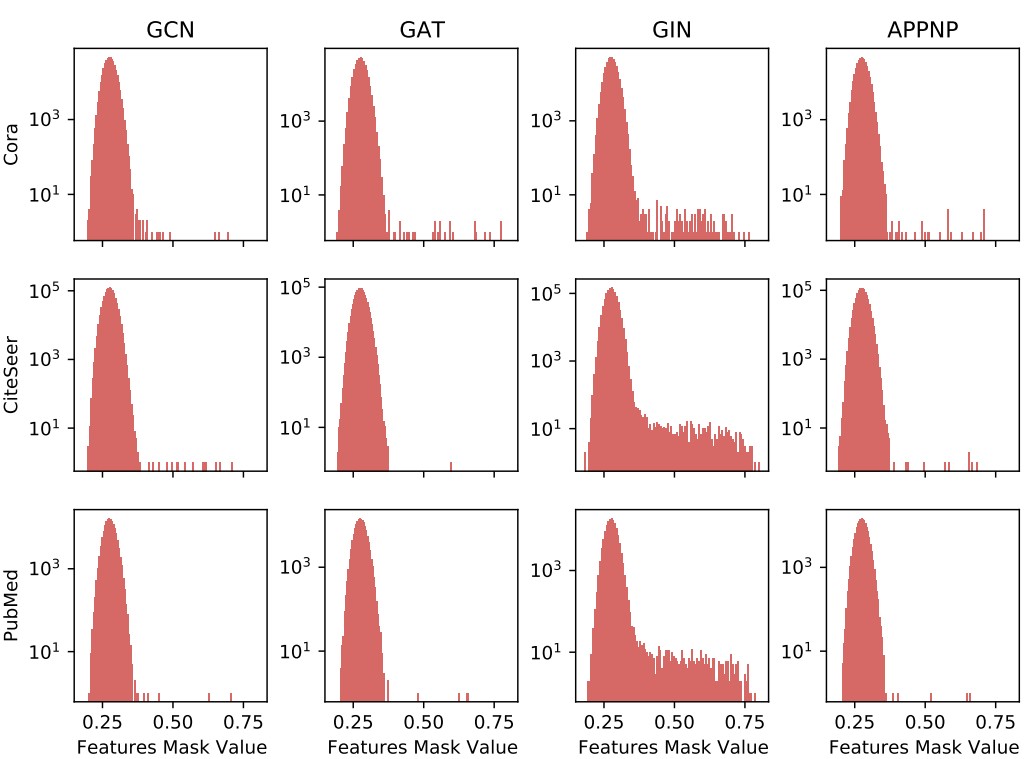

Figure 11: Distribution of GNNEXPLAINER's feature mask values corresponding to explanations of all explained nodes.

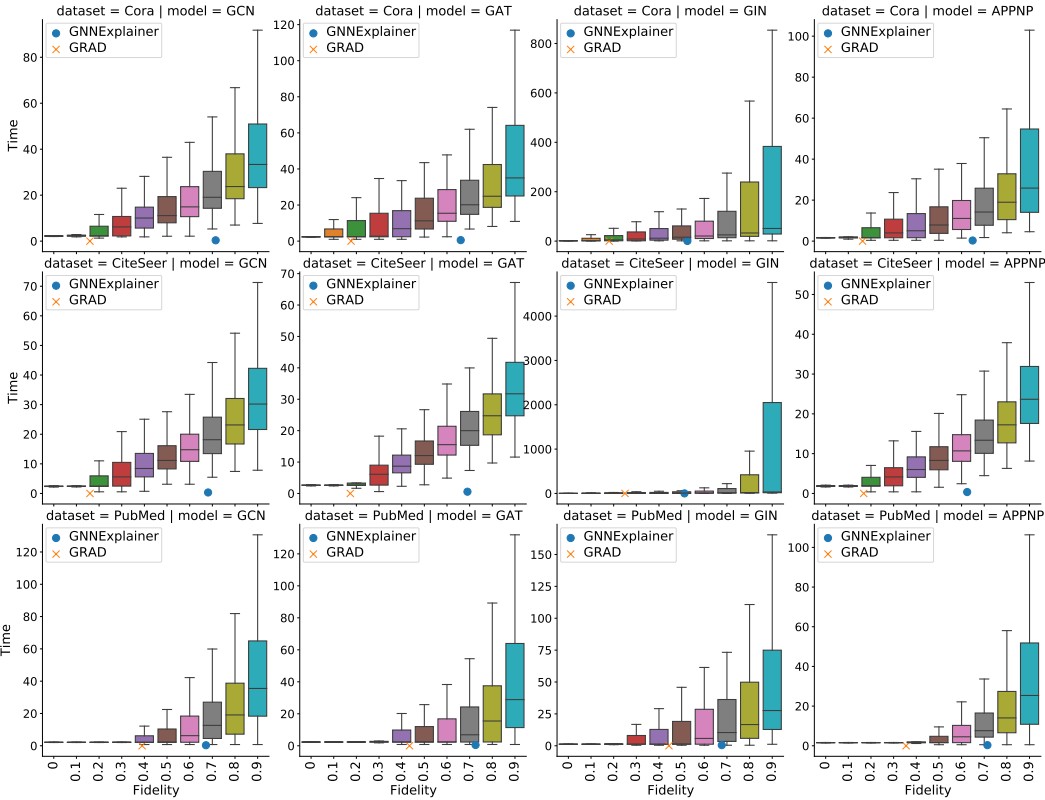

Figure 12: Comparison of the runtime by model and dataset. The cirlce and the cross mark the average fidelity and average time of GNNEXPLAINER and gradient baseline. The boxplots correspond to ZORRO and visualize the runtime for retrieving the first explanation, which has at least the given fidelity. For ZORRO, we recorded the time after the initilization, which can be pre-computed for all nodes simultaneously. Time is measured in seconds.

simultaneously. Limiting the maximum number of features or neighbors would additionally reduce the runtime because the outliers would be avoided.

## E  EXPERIMENTS ON SYNTHETIC DATASET

The synthetic dataset is generated by generating two communities consisting of house motifs attached to BA graphs. Each node has eight feature values drawn from $\mathcal{N}(0,1)$ and two features drawn from $\mathcal{N}(-1,.5)$ for nodes of the first community or $\mathcal{N}(1,.5)$ otherwise. In addition, to follow the published implementation of GNNExplainer, the feature values are normalized within each community, and within each community, $0.01\,\%$ of the edges are randomly perturbed.

The eight labels are given by the following: for each community, the nodes of the BA graph form a class, the 'basis' of the house forms a class, the 'upper' nodes form a class, and the rooftop is a class. The used model is a three-layer GCN, which stacks each layer's latent representation and uses a linear layer to make the final prediction. The training set includes 80% of the nodes.

Since GNNEXPLAINER only returns soft edge mask, we sorted them and added both nodes from the highest-ranked edges until at least five nodes were selected. In this way, we retrieved hard node masks, which are necessary to compare with the ground truth.

To highlight this task's insufficient design, we added as a very simple baseline, a *random closest neighborhood* heuristic, which randomly selects nodes from the nearest neighborhood. For example, if a node has two direct neighbors and 15 nodes in the second neighborhood, we select the two immediate neighbors and sample another three randomly from the second neighborhood.

For the synthetic dataset, we know how many features are not only randomly distributed but contain information about the community. Therefore, we selected the two features correlated with nodes' community membership as ground truth and evaluated the methods' performance with respect to their feature selection. For GNNEXPLAINER, we selected from the calculated soft feature mask those two features with the highest value. In contrast, ZORRO directly select this number during inference. Table 6 shows that GNNEXPLAINER fails to select the informative features and that ZORRO consistently selects the informative features. The configuration with a lower threshold ($\tau = .85$) shows that we only need one of the two informative features in most cases. However, to reach a higher threshold of $\tau = .98$, more features are required.

Table 7 shows how connected the retrieved explanation is. In other words, how close is the explanation to a connected subgraph? We measure this by counting the number of connected components in the explanation with and without the explained node. As we can see, even though in GNNEX-PLAINER the number of connected components is limited to four or three in the case with respectively without the explained nodes. This is the case since we select the nodes based on edges, i.e., each connected component consists of at least two nodes. The explanation tends to be way more disconnected than those of ZORRO.

Table 6: Performance of the feature masks on the synthetic dataset

| Method | # Features | Recall | Precision |
|---|---|---|---|
| Zorro ($\tau = .85$) | 1.48 | **0.98** | 0.68 |
| Zorro ($\tau = .98$) | 2.21 | 0.94 | **0.88** |
| GNNEXPLAINER | 2.00 | 0.08 | 0.08 |

Table 7: Number of the connected components in the explanations of the synthetic dataset. Components$_{without}$ is the number of connected components in the subgraph induced by the selected nodes. # Components$_{with}$ is same, only with the explained node added to the selected nodes.

| Method | | # Components$_{with}$ | # Components$_{without}$ |
|---|---|---|---|
| Zorro ($\tau = .85$) | mean | 1.14 | 1.06 |
| | std | 0.50 | 0.68 |
| Zorro ($\tau = .98$) | mean | 1.27 | 1.35 |
| | std | 1.14 | 1.22 |
| GNNEXPLAINER | mean | 1.73 | 1.56 |
| | std | 0.85 | 0.64 |

To exemplify the performance of our method, the Figure 14 and Figure 15 show some examples of found explanations for correctly respectively wrongly predicted nodes. We compare the explanation of ZORRO against the baseline GNNEXPLAINER. The first node in Figure 14 shows that ZORRO finds multiple explanations, which correspond to the ground truth motif. In contrast, GN-NEXPLAINER ranks four nodes from the BA graph among the highest, which should not affect the prediction of the GCN. The illustrations in c) and d) show a similar pattern. In addition, we see in c) the reason for a low recall of ZORRO. Often only 2-3 nodes of the ground truth are sufficient to reach a high fidelity.

Figure 15 shows some explanations of wrongly predicted nodes. The first node 307 is not in the training set, and from ZORRO's explanation, we get some reasoning about the prediction. The GCN only needs the value from the node itself and two neighbors, which are (wrongly) predicted as the "basis" of the house. Hence, the node is predicted as the top wall, which would follow the following pattern: If from its first neighborhood and its second neighborhood each a node is the basis, then the node is a top wall node. However, both neighbors, which are important for the prediction, were predicted false, and hence the resulting prediction is also wrong. This observation agrees with the observed "homophily of wrong predictions" of the real datasets. GNNEXPLAINER's explanation of the same node is way larger and includes nodes from the house graph of the second community.

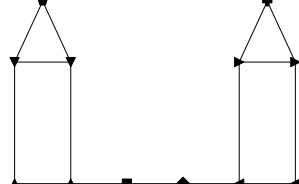

Figure 13: Legend showing the symbols for the different classes of the synthetic dataset. Left and right the house graphs and both nodes (square and rotated square) represent the nodes in the BA graphs of the two communities.

The second example in Figure 15 shows an example, where the explanations of ZORRO and GN-NEXPLAINER are more similar. Both methods retrieve (mostly) members of the ground truth as an explanation for the false prediction.

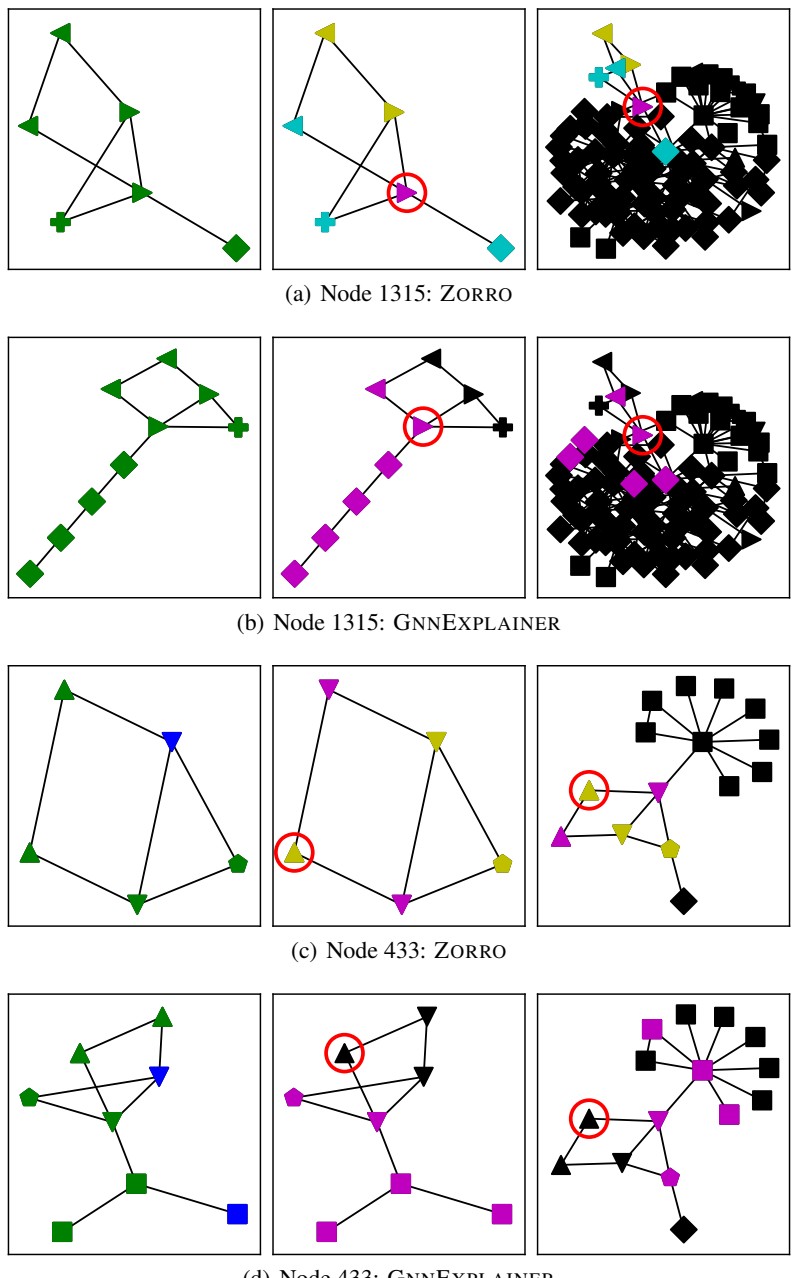

(a) Node 1315: ZORRO

(b) Node 1315: GNNEXPLAINER

(c) Node 433: ZORRO

(d) Node 433: GNNEXPLAINER

Figure 14: Two nodes of the synthetic example explained by ZORRO and GNNEXPLAINER, which are correctly predicted. The first column shows the nodes with the ground truth labels. All nodes from the ground truth (house graph), as well as all the nodes selected in the explanation colored as green (training set) and blue (test set). The second column shows the predicted classes and the explanation(s), where we included in black all unselected nodes of the ground truth. All other colors correspond to each explanation, e.g., three disjoint explanations in a). The red circle highlights the node, which is explained. The last column shows the computational graph of the explained node again with the ground truth labels.

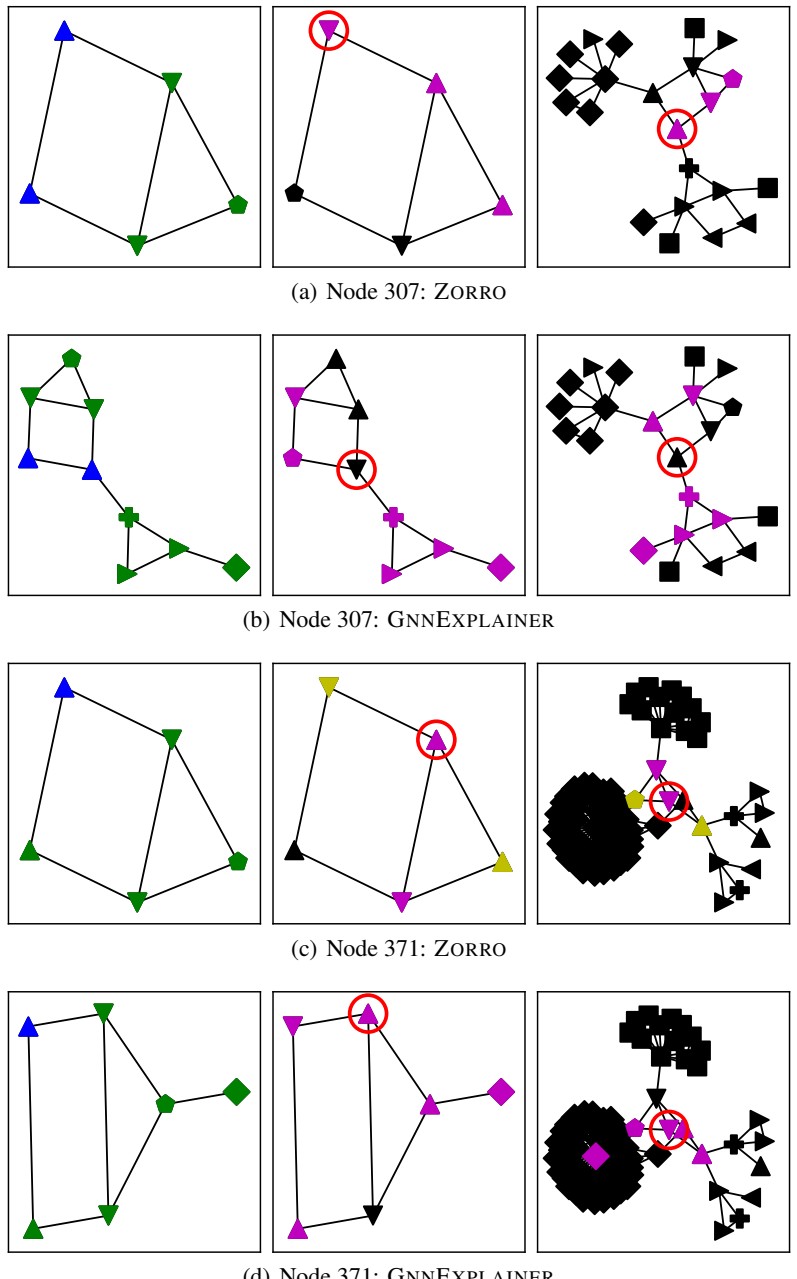

(a) Node 307: ZORRO

(b) Node 307: GNNEXPLAINER

(c) Node 371: ZORRO

(d) Node 371: GNNEXPLAINER

Figure 15: Two nodes of the synthetic example explained by ZORRO and GNNEXPLAINER, which are wrongly predicted. Style same as Figure 14.

