# OpenReview forum: "Hard Masking for Explaining Graph Neural Networks"
_ICLR.cc/2021/Conference — Reject_

### Official Review · AnonReviewer3 · 2020-10-26
**Method for explaining GNN behaviour with room for improvement**

**Rating:** 5
**Confidence:** 4

**Review:**

The authors address the problem of explaining the behaviour of graph neural networks (which operate on a computation graph based on their k-hop neighbourhood) such as a graph convolutional network (GCN).

The core idea is to identify, for each node v in the graph, the nodes and features of the graph most relevant to the behaviour of the GNN for node v. That is, the goal is to find a subgraph of the computation graph associated with a node v in the graph. Importantly, the authors propose to test whether the chosen subgraph is relevant (and the complement wrt the computation graph irrelevant) by adding random noise on the parts deemed irrelevant by their method.

The method is evaluated through a metric called fidelity, which is the agreement in label output between the behaviour of the original and masked GNN, in expectation over the noise distribution.

While overall a well-written paper, a source of confusion is the authors tendency to conflate the computation graph and the graph to which the GNN is applied. The most important notion here is S, defined as a subset of the computational graph. When defining this, it is important to also define precisely this computation graph. What does it look like (abstractly, independent of the GNN instance used)? For instance, there is a nice compact way to unify most message-passing neural networks. (see e.g., https://pytorch-geometric.readthedocs.io/en/latest/notes/create_gnn.html)
When you look at this definition, you see that there are several learnable functions (phi, etc.), the aggregation function, and finally the classification layer. Now, in your definition of a computation graph, what are the nodes? Are the applications of the learnable functions each a node? What about the aggregation (one node?). Again, I think for the reader to fully understand how your explanations S look like, this needs to be rigorously defined. My assumption here was that the computation graph groups computations such that nodes in the computation graph and nodes in the graph to which the GNN is applied coincide. Generally, I think you should spend more effort on section 3. The notation in the argmax statements in section 3.1 is also strange. For instance, S is defined as a pair. So it should be written as argmax F((V_p, {f})). Also, what is the p here?

Another worry I have is the efficiency of the approach. If your average number of features and nodes that exceed the fidelity threshold is K (the average size of S) and the graph has N nodes and F features, you need to evaluate the GNN KN+KF times to obtain an explanation for one node. For large graphs and/or graphs with numerous features, this can be expensive. And this is for the case when you compute the expectation with one Monte-Carlo sample from the noise distribution.

The most disappointing aspect of the paper, however, is the experimental evaluation. Sure it is interesting to assess the multiplicity and size of explanations. What would be more interesting, however, is to evaluate how faithful your explanations really are. And here is where I have a disagreement with your assumptions. You write “[...] that is completely faithful to the model i.e., the explanation achieves the fidelity value of 1.” But achieving a fidelity of 1 does not mean that your explanation is faithful. We could only know this if you removed the nodes and features and retrained the model with the same seeds/initialization. There is an intricate interplay between the nodes and features during training of a GNN. What you evaluate is how close to the original behaviour the GNN is when you remove certain nodes and features. But I would question whether this is a proper definition of faithfulness.

My suggestion would be to also run experiments where you retrain GNNs and check whether the behaviour is indeed such that removing the nodes and features your method deems unimportant leads to a minor change in behaviour.

The synthetic experiments of the GNNExplainer paper are not included. But it makes sense to me to define synthetic graph classes where the presence of certain features/nodes is known to cause the node label by construction. This way one can check whether those features are the ones identified by the XAI method. I would encourage the authors to also run these experiments and compare to the results from the GNN explainer.

Finally, it is not entirely fair to compare other methods to yours through the notion of fidelity alone. Your method is defined to optimize for it. As I mentioned above, fidelity is one way to measure the quality of a reduced graph but not the only one. It is by no means the only one to measure “faithfulness,” as I have outlined above. For instance, it would also make sense to compare based on the measures introduced in the GNNExplainer paper.

---

> ### Author Response · Authors · 2020-11-19
> **Response to Reviewer #3**
>
> Thanks for your comments. For the points raised we have the following response:
> 1) We have now added an abstract formulation of GNNs in Section 3.1. In Section 3.2, we have improved the description of the used computational graph. We have made the observation more explicit that for a k-layer GNN, the information used to create a prediction is contained in the k-hop neighborhood of the query node.
> S Is the set of the selected set of nodes (from the computational graph) and features as an explanation. We have added additional information about the notation in Section 3.2 and also updated the pseudocode accordingly. In addition, we added a table in appendix A with all details (repeated) of the used notation.
> 2) Since this paper's focus was on showing the effectiveness of our approach, we did not invest in making ZORRO efficient. Our methods' computational complexity depends on the number of samples we use to estimate the fidelity. We can make our experiments arbitrarily faster by reducing the number of samples. We also intend to use batching and reusing samples to further improve Zorro's efficiency in the future. We have added a section in the appendix stating the reasoning for our design choices, which already make our current implementation reasonably efficient. We have added complexity analysis in the current version of the article.
> 3) Your suggestion is akin to using Zorro as a feature selection method to train a sparse model. And if the sparse model makes the same prediction as the original GNN, then the explanation is correct.  If we remove the nodes and re-train them, we would end up with a different GNN to measure fidelity. Be that as it may, different nodes have different explanations. A node could be present in one explanation but masked-out in another. This complicates building a valid graph from the explanation masks let alone training.
> But following your suggestion, we retrieved the explanations for all training nodes of the GNN on Cora and selected the top k features, which were most often in the first explanation. Similarly, we retrieved all explanations with GNNExplainer and selected the top k features with the highest summed feature mask values. The results are the following (see Section 4.4):
> | Method              | k=1   &nbsp;| k=10 &nbsp;| k=50 &nbsp;| k=100 &nbsp;|
> |---------------------|-------|------|------|-------|
> | ZORRO ($\tau= .85$) | 0.24* | 0.50* | 0.71* | 0.77* | 0.78* |
> | GNNExplainer | 0.15 | 0.21 | 0.35 |  0.54 |  0.66 |
> (Explained GCN: 0.79, * marks highest value)
> 4) We have now added experiments on synthetic graphs where ZORRO outperforms baseline. Our explanations also highlight the fact the GNN may not require all nodes of the ground truth. To show this, we have added some example explanations in the appendix.
> We believe that fidelity, as defined in the paper, is a more realistic measure of the goodness of explanation as it measures how using the explanation alone leads to the same model decision. Despite that, for synthetic datasets, we use measures other than fidelity as also used by GNNexplainer. Our method also outperforms GNNexplainer under those measures.
> | Method | #Nodes &nbsp;| Recall &nbsp;| Precision &nbsp;| Accuracy &nbsp;|
> |--------|--------|--------|-----------|-----------|
> | ZORRO ($\tau=0.85$) |  2.48 | 0.35 | 0.94* | 0.90* |
> | ZORRO ($\tau=0.98$) | 5.42 | 0.50* | 0.90 | 0.90 |
> | GNNExplainer | 5.34 | 0.35 | 0.33 | 0.79 |
> (* marks highest value, all average values)

---

> > ### Comment · AnonReviewer3 · 2020-11-20
> > **re: response to reviewer 3**
> >
> > Dear authors,
> >
> > Thank you for your response. I have looked at the updated submission and appreciate the improvements I see there. For instance, adding the background on GNNs is beneficial. Here, however, I would also like to press a bit more the issue of making clear what you mean by "computation graph." You write in section 3.2
> >
> > > We note that the computational graph of a node n operation as specified by neighborhood aggregation operation, see Eq. (1), fully determines the information used by GNN to predict its class.  In particular, for a L-layer GNN, the L-hop neighborhood of n will constitute its computational graph.
> >
> > But a computational graph has as nodes usually operations (or a collection of operations, summarized as a "layer"). In your GNN definitions, you use (\ell) for the depth. Intuitively, \ell here correspond to nodes (layers) in the computation graph, no? So if that's not what you mean (and that's what I understand), you might not want to call the L-hop neighborhood the computation graph. It's (as far as I understand it) the subgraph of the input graph that is induced by all nodes that partake in the computation graph for a particular node. This is subtle but it might be confusing for some readers if you equate the L-hop neighborhood of a node and the computation graph that a GNN induces for a particular node. I hope I am making sense.
> >
> > Regarding the efficiency. Sure, you didn't investigate this since you focused on effectiveness. My point is that you should actually look at the efficiency of your approach and include experiments here. What you do is essentially a form of a perturbation based local explanation method. These are known to be expensive. A comparison in runtime to gradient based methods (e.g. integrated gradients) would be nice.
> >
> > Lastly, I still do not agree with the way that you use the term "faithfulness". Your method generates local explanations. There are multiple issues with local methods. In fact, in your experiments you find that there are multiple alternative ways to explain the same prediction. The notion of faithfulness is a strong one. It says that "exactly the presence of these features/nodes caused the model to behave that way". It is very difficult if not impossible to achieve this with local methods. So, I would ask you to not equate fidelity and faithfulness but to consider that fidelity is but one way of measuring whether local explanations are good at maintaining the original behaviour of the model.

---

> > > ### Author Response · Authors · 2020-11-22
> > > **re: re: response to reviewer 3**
> > >
> > > Thanks for your reply.
> > >
> > > **Computational Graph:** We hope the following changes will clarify your remaining concerns about our formulation of the computational graph:
> > >
> > > We note that for a particular node, $n$ the subgraph taking part in the computation of neighborhood aggregation operation, fully determines the information used by GNN to predict its class. In particular, for a $L$-layer GCN, this subgraph would be the graph induced on nodes in the $L$-hop neighborhood of $n$. We will call this subgraph the *computational graph* of the query node. We would like to point out that the term computational graph should not be confused with the computational graph of the neural network.
> > >
> > > **Efficiency:** We now include in the Appendix D (Figure 12) the runtime of our approach. After the initialization, we only need few seconds to retrieve additional elements of the explanation.
> > > During the creation of the paper only very limited methods (GNNExplainer) were published, which allow to explain GNNs. Please note that a comparison on gradient-based methods like Integrated Gradients, GradCam, LRP, DeepLift etc is not trivially possible since their extension to variable-size computational graphs is not well understood for node classification tasks (to the best of our knowledge until the submission of this manuscript). We however compare it with simple gradients that was also used in GNNExplainer.
> > >
> > > **Faithfulness:** We first agree that there is no one perfect way to evaluate the quality of explanations. We agree with the reviewer that one cannot be 100% certain if the selected features/nodes are INDEED the explanation. In that sense, there is no fool proof way to check the REAL fidelity of an explanation. However, we can approximate the actual fidelity by expected fidelity (that we propose in this paper). This notion of expected fidelity is based on information theoretic interpretations that means -- "if the explanation is highly predictive in expectation, then it is a high qualitative explanation". Since we fully agree with the reviewer, we will reflect this in our paper where our claims are qualified with caveats. If it makes more sense we can call our measure "expected fidelity". To avoid any confusion, we have rewritten the respective paragraphs with faithful and removed the term.

---

### Official Review · AnonReviewer1 · 2020-10-26
**Official Blind Review #1**

**Rating:** 4
**Confidence:** 4

**Review:**

This work proposes to explain graph neural networks using hard masking techniques. Specifically, it tries to find the node mask $V_s$ and feature mask $F_s$ which can identify the most important information of the input such that the masked information can yield a high fidelity score. This work proposes a greedy method, ZORRO, to explore these hard masks, which can be used as the explanations of the prediction. Experimental results are interesting and promising.

Strengths:
+ The task is very important. GNNs are very popular but they are mostly treated as black-boxes. Interpreting GNNs is still less studied.
+ Compare with GNN-Explainer, this work focuses on using hard masks to explain GNN predictions. It is a reasonable choice since soft masks, which are used in GNN-Explainer, may introduce new semantic meaning or noise to the node representations since these representations are very sensitive.
+ Experimental results are very interesting. First, there exist multiple explanations for the same input graph that they both lead to high fidelity scores. Second, the proposed method can obtain high fidelity scores than GNN-Explainer and more sparse explanations. In addition, this works studies several types of GNNs, such as GCN, GAT, GIN, APPNP.

Weaknesses:
- The connection between the proposed method and data compression is not convincing. From my understanding, it belongs to the masked-based interpretation methods, which is widely studied in other domains, such as image and NLP. Then I do not think it is something new from other fields--data compression in information theory.
- In the proposed method, all nodes share the same feature mask $V_s$. Is it a proper choice? Is it possible that different nodes may have different important features? Then probably it is better to not share the $V_s$?
- In the proposed method, the ordering information $R_V$ and $R_F$ are stored. It is computed in the beginning and keep fixed for later steps. However, in the later steps, the algorithm will update the $V_S$ and $F_s$, then why do we use the same ordering information? Top nodes, in the beginning, may not be top any more after some nodes/features selected?
- The method itself is very straightforward, which is a simple greedy algorithm. Then I believe the technical contribution may not reach the bar of ICLR.
- The algorithm is not clearly explained. What’s the meaning of $V_r$, $F_r$, $R_{V_p}$, and $R_{F_p}$, etc.? How are they initialized?
- For the comparisons with GNN-Explainer, we need to see some real examples—explanations for both correct predictions and incorrect predictions. It is not enough to just report numerical numbers.


I am willing to adjust my score if my concerns are properly addressed.

=====Update after rebuttal=====

I have read the authors' rebuttal. However, my concerns are not well addressed.

1. There are a lot mask based methods for interpretation in different domains [1] [2] [3] [4]. Existing methods [1][2][3] are providing post-doc explanations for a pretrained model. I still believe "the connection between the proposed method and data compression is not convincing".

2. I still believe the novelty is limited.

Hence, I am keeping my score unchanged.

[1] GNNExplainer: Generating Explanations for Graph Neural Networks, NIPS 2019

[2] Real Time Image Saliency for Black Box Classifiers, NIPS 2017

[3] Learning to Explain: An Information-Theoretic Perspective on Model Interpretation, ICML 2018

[4] Rationalizing Neural Predictions, EMNLP 2016

---

> ### Author Response · Authors · 2020-11-19
> **Response to Reviewer #1**
>
> Thanks for your review. We discuss the raised weaknesses consecutively and have numbered them to be precise:
>
> 1) The masked-based interpretation methods alluded by the reviewer are predominantly for models that use masking during model building – like in Language (Lei et al. 16, Bastings et al. ‘19, Invase ICLR’19). We are crucially different in that we operate only on the trained model – the post-hoc setting. We have no access to the parameters of the trained model. We only assume that we have access to the final prediction. We also accept that there are parallels to the above masking methods. However, where we crucially differ from existing “masking methods” (although NOT post-hoc) is that we measure the fidelity in expectation by sampling multiple instances. This provides our fidelity estimates necessary robustness that is missing in earlier works. The rate-distortion theory merely provides a theoretical framework on which we ground our fidelity measure.
>
> 2) Our method is a local interpretability method. We explain the class label predicted for a single node. For the explained node, we currently use the same feature mask for all selected neighbors to reduce the explanation's complexity. It would be easy to extend our method such that we search for each node in the computational graph its feature mask.
>
> 3) In short, we do not recalculate the orderings RF and RV because we did not observe any performance gains from doing so and initializing them already has the highest impact on our runtime. We have added a section in the appendix (see Appendix A) explaining the reasoning for our algorithm's design choices.
>
> 4) The contribution is not merely the greedy algorithm, which is simple, as the reviewer rightly points out. Our major contribution is proposing a principled framework under which the validity of GNN models' explanations can be measured. In doing so, we intend to enrich the underexplored area of GNN explainability by proposing our fidelity measure. Also, the insights from our "simple" approach (1) already showcase the limitations of existing approaches [Section 4.1], (2) is empirically superior to the existing approach [sections 4.2, 4.4, 4.5], and (3) shows how explanations can be used to inspect trained models [Section 4.3]. These utility experiments, to the best of our knowledge, have completely been missing in the literature.
>
> 5) We have rewritten the description of our algorithm to make it easier to understand. In addition, we have added a lengthy description of all details in Appendix A.
>
> 6) We now include some example explanations for the synthetic dataset.  See Figure 13 and Figure 14 for correctly and wrongly predicted nodes.

---

### Official Review · AnonReviewer2 · 2020-10-28
**Interesting method - but no empirical evidence of whether explanations are meaningful**

**Rating:** 5
**Confidence:** 4

**Review:**

The authors propose ZORRO, a post-hoc explanation method for node classification with graph neural network architectures. ZORRO leverages rate-distortion theory to generate masks that select nodes in the target node neighbourhood and their most important features.

* The problem is very relevant to the GNN community, and I am glad to see more works coming in on this topic.
* The idea of relying on rate-distortion theory is interesting and original, and to the best of my knowledge this is the first time I see it used to tackle this research problem.
* The paper is well structured and organised.
* The original contribution is sufficient.
* Related work is sufficiently well covered.

Nevertheless, the paper suffers from shortcomings:

A) The paper claims that ZORRO can generate multiple, disjoint explanations, apparently all highly faithful. This seems to be at odds with the authors' claim to explain the behaviour of the model. In other words, if I were on the receiving side and I was given multiple _disjoint_ explanations, which one should I trust more? How can explanations shed a light on the  behaviour of models if they are disjoint? I see ZORRO is able to generate overlapping explanations as well (a property compatible for example with example-based technique in XAI literature such as counterfactual explanations), but I have mixed feelings on the effectiveness of disjoint explanations in practice.

B) A drawback of this work is the complete absence of human-based evaluation. I acknowledge explainable AI literature is ripe with examples of accepted papers, but the authors seem to deliberately disregard this aspect (“We [.. ] are not interested if an explanation is congruent to human understanding", Sec1). If humans are not important, then what is the reason you explain your predictions? If the goal is limiting to debugging a model, perhaps the narrative should be revisited. All in all, I believe users should be central in an XAI piece, and papers in this area should help the reader understand if the generated explanations meet users expectations - even in a ML conference such ICLR, even for a 8-page paper.

C) The paper does not include any examples of the generated explanations. It is hard to figure out if the claimed fidelity brings meaningful results in practice, and the reader is left with this doubt. Aside from a full-fledged evaluation campaign (see A. above), some examples would really help make the case.

D) Experiments do not include any evidence of whether ZORRO explanations work in practice. Besides, as I mentioned above, the author should probably clarify which audience they are targeting (i.e engineers debugging a model, end users trying to understand the reasons for a specific model outcome, etc.)

E) I was expecting experiments to assess the impact of \tau (the user-defined fidelity threshold). The authors experiment with .98 and .85 - and “the choice of \tau has limited influence”, but I would have expected empirical evidence for such statements (i.e. experiments on a wider range of \tau, to assess size and fidelity). If choosing a desired fidelity is not crucial, the paper should show so.

F) There are not experiments on runtime complexity. The reader is left without evidence of how long it takes to generate an explanation for a target prediction.

G) It is unclear if ZORRO achieves more faithful and also smaller explanations than GNNExplainer. Sec 4.2 suffers from clarity issues, as long as Figure 5.

H) Some sections could be better clarified, to help the reader understand important aspects of the work: Example: In sec1, the “notations” paragraph would benefit from proofreading and re-wording. Sec 4.2 could also be refined.

Minor:
* Size matters for explanations, but smaller does not always mean better. For example, in medical decision support systems, some clinicians may prefer longer and more thorough explanations. Your milage may vary.
* Figure 5 is poorly legible.
* Some typos along the way (e.g. “denotes the binary column vector of selected nodes and Fs denote the binary row vector of selected nodes “ sec 1, “Effectively the complete input is presented as an input” in 4.2)

Questions for the authors:
* Q1) How does ZORRO decide the size of each computational graph to work with (i.e. the size of the neighbourhood)?
* Q1) It is not entirely clear to me how you ZORRO generates multiple explanations? I could not find how this is done in Algorithms 1-3. Could you please clarify?

---

> ### Author Response · Authors · 2020-11-19
> **Response to Reviewer #2 (1)**
>
> Thanks for the kind and positive comment. We view our paper as a step towards a better evaluation of post-hoc interpretability approaches for GNNs, and we are pleased about the constructive comments that allow us to reflect on the limitations of the current evaluation regimes.
> Our detailed response to your raised points:
>
> A) We intend to exactly shed light on the multiple-explanation limitation of the existing GNN explanation approaches. Unlike existing approaches that output a SINGLE explanation, we experimentally show that multiple explanations exist with better or similar fidelity as the soft explanation. Therefore, an expectation that there is ONE perfect explanation might indeed be misplaced. There are multiple possible reasons for a given prediction based on the feature and neighborhoods, and outputting only one of them is akin to not making the user aware of the real evidence. We choose to output multiple disjoint explanations as a lower bound on the potential number of explanations that could exist. We expect that there are many overlapping explanations.
> We accept the concern that the beneficiary of multiple explanations would not guide the user to improve the model or localize potential spurious correlations. Still, through this paper, we want to question the assumption (or expectation) that a single explanation provides a complete picture of the inner workings of a GNN.
>
> B) We accept the claim that explanations are meant for users and hence should be evaluated by humans. However, in our opinion, what can be assessed by a human is the explanation style and not the effectiveness of explanations. In this regard, soft masking techniques that output a probability distribution over features or nodes are well-known to be less interpretable to humans than hard masks as explanations. This is especially true when dealing with large number of features as in our experiments (Cora has > 1K features, and some nodes have a very high outdegree).
> However, we would want to point out a central limitation of existing works in the human evaluation of explanation methods. Human evaluation regimes where a human is showed multiple explanations and is asked to choose the best explanation is fraught with multiple biases and is not a true indicator of “goodness of the explanation”. Why? Imagine there are two explanations to a GNN prediction.
> (a) An incorrect explanation that corresponds to human understanding (say Homophily) and
> (b) another correct that does not align with human understanding (a spurious correlation perhaps).
> In such cases, the Human evaluation might incorrectly evaluate the wrong explanation. This has been routinely observed for post-hoc explanation methods in text and images (cite Rigorous Science of Interpretability by Been Kim & Doshi Velez). All post-hoc interpretability approaches face similar threats when using human explanations. Consequently, we propose an information-theoretic point of view: If an explanation is correct, it must be predictive.
>
> C) We have included some examples for the newly added synthetic dataset, which showcase our approach's results. Please take a look at Figures 13 and 14.
>
> D) This a valid comment about the stakeholders that we target. We make it more explicit that our explanations are mainly targeted towards model builders, designers, and practitioners of GNNs. We see Zorro's utility in the evaluation and debugging phase to get insights into the model under question. Our utility experiments showcase one possible way in which Zorro was used to create visualizations that shed some light on the learning behavior of GNNs.

---

> ### Author Response · Authors · 2020-11-19
> **Response to Reviewer #2 (2)**
>
> E) Our goal of choosing different values of $\tau$ is to show that our method can retrieve small-sized explanations at a higher value of $\tau$ too. In principle, $\tau$ is a hyperparameter that a user can specify. The higher the value of $\tau$ better the explanation explains the model behavior.  By construction, our approach will output explanations with fidelity higher or at least equal to $\tau$. Choosing the desired fidelity is crucial. We aimed to illustrate in our experiments with different $\tau$ that for higher fidelity, the explanation size may not necessarily increase. A point to be noted here is that for some nodes, we found that GNNexplainer obtains a fidelity of 1, but a close inspection showed us that feature mask values are so distributed that each feature obtains almost the same importance, implying that the evaluation on soft masked explanation allowed the model to use nearly all of the features. That is why we emphasize checking the change in the size of our explanations for high fidelity (high $\tau$) cases.
>
> F) Since this paper's focus was on showing the effectiveness of our approach, we did not invest in making ZORRO efficient. Our methods' computational complexity depends on the number of samples we use to estimate the fidelity. We can make our experiments arbitrarily faster by reducing the number of samples. We also intend to use batching and reusing samples to further improve Zorro's efficiency in the future. For the new experiments on the synthetic dataset, we recorded on average a runtime way below a minute for each explanation. Since our runtime only depends on the number of nodes in the computational graph, the number of features and the number of samples, the runtime does not necessarily increase with the size of the graph.
>
> G) We have rewritten section 4.2 to make our point clearer: To show that Zorro, in fact, finds smaller explanations than GNNexplainer, we compare the entropy over the normalized feature mask distribution. For Zorro, we have a binary mask, which implies that Zorro's entropy will be equal to the log (number of selected features). Lower entropy here means a smaller number of selected features.
> In addition, we now have included results from the experiments on synthetic data from GNNExplainer, which support the above arguments. The ground truth of the dataset contains five nodes, and ZORRO keeps the explanation size small by choosing on average 2.48/5.42 nodes for $\tau=.85$ resp. $\tau=.98$.
>
> Q1) We choose the same computational graph as the underlying GNN algorithm. Specifically, for a k-layer GNN, we use the node's k-hop neighborhood as the computational graph. We now added a background section of GNNs (section 3.1) and described the used computational graph more clearly in Section 3.2.
>
> Q2) Lines 13 and 14 of Algorithm 3 perform recursive calls to Algorithm 3, which allows retrieving multiple explanations.

---

### Decision · Program_Chairs · 2021-01-07
**Final Decision**

**Decision:**

Reject

**Comment:**

The paper provides a simple approach to explaining GNN predictions for each node by greedily selecting nodes or features in each computation graph so as to increase the fidelity score. The fidelity score is based on comparing the original GNN output to what is obtained with noisy versions of the masked nodes/features. While simple, the approach seems somewhat inefficient (efficiency should be assessed/characterized). Also, several improvements to the evaluation expressed in the reviews/discussion (e.g., human evaluation, practical utility, comparison to gradient based methods) would make the submission somewhat stronger.